# Rolling Bearing Incipient Fault Diagnosis Method Based on Improved Transfer Learning with Hybrid Feature Extraction

**DOI:** 10.3390/s21237894

**Published:** 2021-11-26

**Authors:** Zhengni Yang, Rui Yang, Mengjie Huang

**Affiliations:** 1School of Advanced Technology, Xi’an Jiaotong-Liverpool University, Suzhou 215123, China; Zhengni.Yang19@alumni.xjtlu.edu.cn; 2Institute of Information Technology, Xinjiang Teacher’s College, Urumqi 830043, China; 3Research Institute of Big Data Analytics, Xi’an Jiaotong-Liverpool University, Suzhou 215123, China; 4Design School, Xi’an Jiaotong-Liverpool University, Suzhou 215123, China; mengjie.huang@xjtlu.edu.cn

**Keywords:** bearing fault diagnosis, incipient fault, transfer learning, domain adaptation

## Abstract

Data-driven based rolling bearing fault diagnosis has been widely investigated in recent years. However, in real-world industry scenarios, the collected labeled samples are normally in a different data distribution. Moreover, the features of bearing fault in the early stages are extremely inconspicuous. Due to the above mentioned problems, it is difficult to diagnose the incipient fault under different scenarios by adopting the conventional data-driven methods. Therefore, in this paper a new unsupervised rolling bearing incipient fault diagnosis approach based on transfer learning is proposed, with a novel feature extraction method based on a statistical algorithm, wavelet scattering network, and a stacked auto-encoder network. Then, the geodesic flow kernel algorithm is adopted to align the feature vectors on the Grassmann manifold, and the k-nearest neighbor classifier is used for fault classification. The experiment is conducted based on two bearing datasets, the bearing fault dataset of Case Western Reserve University and the bearing fault dataset of Xi’an Jiaotong University. The experiment results illustrate the effectiveness of the proposed approach on solving the different data distribution and incipient bearing fault diagnosis issues.

## 1. Introduction

Currently, rotating machinery is becoming a critical component of modern industrialisation. With the continuous improvement of the safety and reliability requirements of mechanical equipment, the research on the fault diagnosis of rotating machinery has become indispensable, and rotating machinery fault diagnosis has become a major research direction in machinery fault diagnosis. In rotating machinery, rolling bearings are one of the most critical components, and bearing failures will affect the operation of the equipment and cause the failure of the rotating machinery. Wu et al. [1] pointed out that rolling bearing failures cause approximately 30% of failures in rotating machinery. In recent years, model-driven methods and data-driven methods are the main two methods for bearing fault diagnosis. The model-driven methods can achieve excellent performance, but the high performance heavily relies on accurate models, and developing a high-quality model is costly. Meanwhile, the data-driven methods can be applied in systems where models are not available, and data-driven methods can diagnose bearing faults based on collected vibration signals. In recent years, various methods of bearing fault diagnosis have been developed, among which data-driven methods for bearing fault diagnosis have achieved outstanding performance.

By combining different feature extraction and selection methods, the conventional machine learning based methods have achieved high classification accuracy, and various algorithms including support vector machine [2] and fast-spectral kurtosis filter [3] have been applied to increase classification accuracy through extracting various features. The features of the data are exactly concurrent and automatic without manual selection in deep learning based fault diagnosis methods compared with the conventional machine learning based methods. Therefore, the deep learning based fault diagnosis methods are considered as an efficient and end-to-end learning system due to the advantage of being able to complete the classification in one step from the input of the original signals to the output [4].

Both conventional machine learning based methods and deep learning based methods require a major assumption that the training data and the testing data must follow the same data distribution. However, this assumption cannot be satisfied in the practical application. As a result, while the classifier performs well on datasets with the same statistical distribution, it struggles to achieve acceptable classification accuracy on datasets with different statistical distribution.

Transfer learning (TL) based methods, in contrast to deep learning and conventional machine learning based methods, can reuse prior knowledge or a developed model to address new tasks of a target domain with different statistical distributions by minimizing the difference between different domains [5]. As demonstrated [6,7,8], the transfer learning based methods have achieved excellent performance in fault diagnosis and image classification.

In addition to statistical distribution requirements, quality and plenty of labeled training data are also required conditions for conventional machine learning and deep learning to train classifiers or models [9]. However, in many practical scenarios, labeled training data is insufficient or costly [10]. It is extremely difficult to gather enough labeled faulty data in rolling bearing fault diagnosis because vibration signals obtained from the same type of fault but under various working conditions can have distinct data distributions [11]. The features of rolling bearing fault in the early stages are inconspicuous to identify and the features extracted from the vibration signal of a later stage may have a different statistical distribution from that of early stage. To solve the features that are inconspicuous of the early stage, Saidi [12] proposed an empirical mode decomposition based approach. The approach decomposes the vibration signal of the early stage into a number of static intrinsic mode functions based on the local characteristic time scale of the signal to identify, and the approach achieved excellent performance on the dataset with the same data distribution. Overall, the insufficient labeled faulty data, imbalanced data distribution and inconspicuous feature of incipient bearing fault eventually result in a significant barrier in bearing fault diagnosis.

In this paper, an incipient fault diagnosis approach based on transfer learning is proposed to solve the above mentioned problems. The proposed approach employs the correlation alignment (CORAL) algorithm to deal with the data distribution problem in different domains, as well as a hybrid feature extraction algorithm based on statistical algorithms, stacked auto-encoder (SAE) network, and wavelet scattering network. The sigmoid entropy is applied on feature vectors after feature extraction to construct feature matrices of both domains in the proposed approach. The feature matrices are aligned on the manifold space by taking the geodesic flow kernel (GFK) method. The k-nearest neighbor (KNN) algorithm is used to classify the faulty data of the target domain for fault diagnosis. The proposed approach is an unsupervised method for rolling bearing incipient fault diagnosis, which does not require any labeled data of the target domain. The proposed approach only uses the labeled data of the source domain to train the classifier, which means the proposed approach is practical. To verify the effectiveness of the proposed methods in fault diagnosis, experiments are conducted on two rolling bearing datasets, the Case Western Reserve University (CWRU) dataset and the Xi’an Jiaotong University-Sumyoung (XJTU-SY) dataset. By comparison, the proposed approach has achieved an outstanding performance among several approaches.

The main contributions of this paper are: (1) a transfer learning based unsupervised incipient fault diagnosis approach is proposed; (2) a novel hybrid feature extraction algorithm based on a statistical algorithm, SAE network and wavelet scattering network is proposed. The rest of this paper is organized as follows: Section 2 introduces the basic concepts and the proposed algorithm, Section 3 discusses the datasets used in this study and the experiment results, and Section 4 concludes this paper.

## 2. Preliminaries and Methods

To deal with the different data distribution issues associated with bearing incipient fault diagnosis, the structure of the proposed approach is illustrated in Figure 1. The CORAL algorithm is used to align the original data of the source domain and target domain in the proposed approach first. Then the hybrid feature extraction method is adopted on both domains and the sigmoid entropy function is applied on feature vector matrices. Lastly, two feature vector matrices are aligned on the Grassmann manifold space using the GFK method, and the faulty samples of the target domain are diagnosed using the KNN classifier trained with the source domain.

### 2.1. Domain Adaptation Algorithm CORAL

The domain adaptation method used in the proposed approach is the CORAL algorithm, which is unsupervised and can minimize the difference between domains. The CORAL is a simple yet effective method for unsupervised domain adaptation, and the effectiveness of the CORAL algorithm has been demonstrated by Sun [13]. The CORAL algorithm is more straightforward and has a lower computation cost than other domain adaptation algorithms [14,15,16].

In the CORAL algorithm, it is supposed that the data $x\in R^{d}$ is the source domain and the data $u\in R^{d}$ is the target domain, both *x* and *u* are d-dimensional, $D_{s}\;=\;x_{i}$ represents the data of the source domain, $L_{s}\;=\;y_{i},y\in \left\{1,\ldots ,L\right\}$ represents the labels of the source domain, and $D_{t}\;=\;u_{i}$ represents the data of the target domain. It is supposed that $C_{S}$ represents the covariance matrices of the source domain and $C_{T}$ represents the covariance matrices of the target domain ($C_{S}\;\neq \;C_{T}$). The CORAL algorithm applies a linear transformation A to the source domain and uses the Frobenius norm as the matrix distance metric to minimize the covariance distance of the source domain and target domain; the equation is shown as below.
(1)$$\min_{A}{∥C_{\hat{S}}-C_{T}∥}_{F}^{2}\;=\;\min_{A}{∥A^{⊤}C_{S}A-C_{T}∥}_{F}^{2}$$
where $C_{\hat{s}}$ represents the covariance of the transformed source features, and ${∥\cdot ∥}_{F}^{2}$ represents the squared matrix Frobenius norm [13].

The Algorithm 1 of CORAL is shown as below:
**Algorithm 1** CORAL for Unsupervised Domain Adaptation**Input:** Data of Source Domain $D_{S}$, Data of Target Domain $D_{T}$**Output:** Data of Adjusted Source $D_{s}^{*}$$C_{S}$ = $cov\left(D_{S}\right)\;+\;eye\left(size\left(D_{S},2\right)\right)$$C_{T}$ = $cov\left(D_{T}\right)\;+\;eye\left(size\left(D_{T},2\right)\right)$$D_{S}$ = $D_{S}*C_{S}^{\frac{-1}{2}}$$D_{S}^{*}$ = $D_{S}*C_{T}^{\frac{1}{2}}$


The process of the CORAL algorithm is illustrated in Figure 2 [13]. Figure 2a shows the different original data distributions of the source domain and target domain. Figure 2b illustrates the data distribution of source domain after decorrelation. Figure 2c shows the data distribution of the source domain after re-correlation using covariance of the target domain.

### 2.2. Feature Extraction

In the fault diagnosis problem, the design of feature extraction plays a significant role in obtaining a high accuracy diagnosis result. In conventional machine learning, the feature extraction of signals is commonly based on the time domain, frequency domain and time-frequency domain [17]. Compared with conventional machine learning methods, the deep learning methods adopt a possibly complex learning system to extract deep features. In recent years, the deep learning methods perform well on image, text classification, and speech recognition [18,19,20]. In this paper, a hybrid feature extraction method is proposed based on both deep learning and conventional feature extraction algorithms are applied on aligned domain datasets, with the structure shown in Figure 3.

#### 2.2.1. Time and Frequency Domain Analysis

In data-driven rolling bearing fault diagnosis, one of the common strategies to extract process parameters is to analyze the mechanical vibration signals with statistical signal processing techniques [21,22]. The statistical features normally indicate the feature of time domain, frequency domain and time-frequency domain. In this paper, ten types of time-domain features are extracted: mean, standard deviation, variance, peak-peak value, root mean square, waveform factor, crest factor, impulse factor, kurtosis, and skewness.

In the frequency-domain, the Fast Fourier Transform (FFT) is applied to extract the frequency-domain characteristic indicators. The FFT algorithm is an efficient method to transfer a time series into a frequency-domain representation. In this paper, four types of frequency-domain features are extracted: center-of-gravity frequency, power spectrum, frequency variance, and mean square frequency. The FFT of a signal can be computed using the following equations:(2)$$\mathrm{FFT}\left(k\right)\;=\;\sum_{n=1}^{N}x\left(j\right)\omega_{N}^{\left(n-1)(k-1\right)}$$
(3)$$\omega_{N}\;=\;e^{(-2\pi i)/N}$$
where *N* is the number of samples for one signal and ωN is an *N*th root of unity [23].

In the time-frequency-domain, the wavelet packet decomposition (WPD) is used to extract the time-frequency domain features in this paper [24]. WPD is an efficient method multi-resolution to decompose the signal into high-frequency part A and low-frequency part D. Thus, an in-depth decomposition for both high-frequency and low-frequency bands can be obtained. The proposed approach extracts the coefficients of 15 nodes on level 4 decomposition as time-frequency domain features as shown in Figure 4.

#### 2.2.2. Wavelet Scattering Network

With a wavelet convolution layer, non-linear layer, and pooling layer, the data is filtered using predefined wavelet and scaling filters in a wavelet scattering network [25]. The structure of the wavelet scattering network is shown in Figure 5.

In Figure 5, the original signal in layer 0 is averaged with a low-pass filter. Afterwards, the discarded high frequency details can be re-captured in subsequent layers by performing a continuous wavelet transform on the input signal to generate a set of scale map coefficients. The nonlinear operator and the low-pass filter are adopted to generate a set of scattering coefficients of layer 1. Then the scale map coefficient output of the previous layer is used as the input of the next layer in the network. Thus, the same nonlinear operator and the wavelet low-pass function can be applied to filter the output to obtain the scattering coefficients of layer 2.

#### 2.2.3. Stacked Auto-Encoder Network

The auto-encoder network is an unsupervised learning method, which takes the backpropagation algorithm to ensure the output data is equal to the input data. The core idea of the auto-encoder network is trying to learn a function $h_{W,b}\left(x\right)\;\approx \;x$ to approximate an identity function, so the output $Y_{i}$, i∈{1,2,3,…,i} is as close as to the input $X_{i}$, i∈{1,2,3,…,i} as possible.

In this paper, the SAE network consists of two auto-encoders, and the structure of the SAE network is shown in Figure 6. The SAE network takes the aligned data of both domains as input data, then the output of the hidden layer in the first auto-encoder is taken as the input of the second auto-encoder. Finally the output of the hidden layer of the second auto-encoder is extracted as features.

The cost function in training the auto-encoder network consists of three terms: the mean squared error term, the *L*_2_ regularization term, and the sparsity regularization term, as shown below:(4)$$E\;=\;\underset{\mathrm{mean}\;\mathrm{squared}\;\mathrm{error}\;}{\underset{︸}{\frac{1}{N}\sum_{n=1}^{N}\sum_{k=1}^{K}\left(x_{kn}-{\hat{x}}_{kn}\right)}}\;+\;\lambda *\underset{L_{2}}{\underset{︸}{\Omega_{\mathrm{weight}}}}\;+\;\beta *\underset{\mathrm{sparsity}}{\underset{︸}{\Omega_{\mathrm{sparsity}}}}$$
where $\Omega_{\mathrm{weights}}\;=\;\frac{1}{2}\sum_{l}^{L}\sum_{j}^{n}\sum_{i}^{k}{\left(w_{ji}^{\left(l\right)}\right)}^{2}$ is the $L_{2}$ regularization term, $\Omega_{\mathrm{sparsity}\;}=\;\sum_{i=1}^{D}KL\left(\rho ∥{\hat{\rho }}_{i}\right)$ is the sparsity regularization term, λ is the coefficient for the $L_{2}$ regularization term, β is the coefficient for the sparsity regularization term, *L* is the number of hidden layers, *n* is the number of examples, *k* is the number of variables in the training data, *D* is the number of neurons in the hidden layer, KL is the Kullback-Leibler divergence to measure the difference between two distributions, ρ is a sparsity parameter, typically a small value close to zero, and ${\hat{\rho }}_{i}$ is the average activation of hidden unit *i*.

### 2.3. Geodesic Flow Kernel

In this paper, all feature vectors of both domains are aligned on manifold with the GFK algorithm. The core idea of GFK is to construct a geodesic curve Φ(t) between the two points on a manifold, then integrate them along the geodesic curve. The mapping in the manifold is shown in Figure 7, where the data of the source domain and the target domain are mapped from the original space into the subspace manifold, and the source domain is transformed into point *S* and the target domain is transformed into point *T* on the manifold. The distribution of three example classes represented by a circle, triangle, and square can be adjusted through the measurement of geodesic flow, represented by the green curve in Figure 7. Concretely, raw features of both domains are projected on these subspaces to form the feature vector [16]. Then the kernel function can be computed in the closed-form on the original feature space based on the feature vector inner product. Thus, the low-dimensional representation of the domain can be derived by using this kernel in the learning algorithm.

For geodesic flow, the subspace angle between the source domain and the target domain is computed first. It is supposed that $D_{S}$ is the source domain and $D_{T}$ is the target domain; first the subspaces $P_{S}$ of the source domain and $P_{T}$ of the target domain (where $P_{S}$, $P_{T}\in R^{D\times d}$) are identified by computing the principal components analysis of two domains. Let $R_{S}\in R^{D\times \left(D-d\right)}$ indicates the orthogonal complement to $P_{S}$ (where $R_{S}^{T}P_{S}$ = 0). Using the canonical euclidean metric on the Riemannian manifold, the geodesic flow is parameterized as Φ:t∈[0,1]→Φ(t)∈G(d,D) under constraints $\Phi \left(0\right)\;=\;P_{S}$ and $\Phi \left(1\right)\;=\;P_{T}$ [16] as
(5)$$\Phi \left(t\right)\;=\;P_{S}U_{1}\Gamma \left(t\right)\;-\;R_{S}U_{2}\Sigma \left(t\right)$$
where $U_{1}\in R^{d\times d}$ and $U_{2}\in R^{(D-d)\times d}$ are orthonormal matrices.

The orthonormal matrices are given by the following pair of singular value decompositions as follows:(6)$$P_{S}^{T}P_{T}\;=\;U_{1}\Gamma V^{T},\;R_{S}^{T}P_{T}\;=\;-U_{2}\Sigma V^{T}$$
where Γ and Σ are d × d diagonal matrices.

The diagonal elements are $\cos\theta_{i}$ and $\sin\theta_{i}$ for i = 1, 2, …, d where $\theta_{i}$ represents the principal angles between $P_{S}$ and $P_{T}$:(7)$$0\;\leq \;\theta_{1}\;\leq \;\theta_{2}\;\leq \;\cdots \;\leq \;\theta_{d}\;\leq \;\pi /2$$
where $\theta_{i}$ is used to measure the degree of overlapping subspaces, and Γ(t) and Σ(t) are diagonal matrices whose elements are $\cos(t\theta_{i})$ and $\sin(t\theta_{i})$, respectively.

The geodesic flow kernel can be computed based on the geodesic flow. For two original feature vectors $x_{i}$ and $x_{j}$, project the original feature vectors into Φ(t) for continuous *t* from 0 to 1 and concatenate all the projections into infinite-dimensional feature vectors $z_{i}^{\infty }$ and $z_{j}^{\infty }$. Thus, the inner product between $z_{i}^{\infty }$ and $z_{j}^{\infty }$ defines the geodesic flow kernel as below:(8)$$\begin{array}{cc} \langle z_{i}^{\infty },z_{j}^{\infty }\rangle & =\;\int_{0}^{1}{\left(\Phi {\left(t\right)}^{T}x_{i}\right)}^{T}\left(\Phi {\left(t\right)}^{T}x_{j|}\right)dt\\ & =\;x_{i}^{T}Gx_{j} \end{array}$$
where $G\in R^{D\times D}$ is a positive semidefinite matrix.

### 2.4. K Nearest Neighbor Classification

In this paper, the KNN algorithm is adopted as the rolling bearing fault identification method. The core idea of KNN is that if most of the k nearest samples in the feature space of a sample belong to a certain category, the sample also belongs to this category. As a non-parametric learning algorithm, the KNN algorithm classifies new samples using a predefined distance measure based on the available information of stored samples.

## 3. Experiments and Results

### 3.1. Data Description

In this paper, six experiments are carried out to verify the effectiveness of the proposed approach. The CWRU bearing fault dataset [26] and XJTU-SY bearing fault dataset [27] are chosen as source domain and target domain, respectively. In this paper, the source domain consists of 600 labeled samples from the CWRU dataset, including 500 vibration signals in each sample. The fault diameter is 0.007 inches and each sample was recorded on a 1797 rpm working condition. Meanwhile, the target domain is made up of 900 unlabeled incipient fault samples from the XJTU-SY dataset, each sample contains 500 vibration signals, and were recorded on a 2250 rpm working condition. All the vibration signals of incipient fault samples were recorded at first 1.28 s during the rolling bearing run-to-failure experiment of the XJTU-SY test stand. Table 1 illustrates more information of the domains. Two types of faults, the inner and outer race wearing, are considered in this experiment and the time-domain waveform of original vibration signals of these two faults and non-fault situation are shown in Figure 8.

### 3.2. Experiment Results and Analysis

In the proposed approach, the CORAL algorithm is adopted to align the statistical distributions of the source and target domains initially. Then the feature extraction methods are adopted to aligned the data to extract 158 features each from the source and target domain. Afterwards, the value of sigmoid entropy is computed based on the extracted features. Finally, the GFK method is used to align the feature vectors of both domains on a manifold, and the K-NN classifier with K = 5 is used to classify the samples of the target domain for incipient fault diagnosis, with K = 5 trained on the data of the source domain. In order to verify the effectiveness of the proposed approach, the proposed method is set as the baseline and five variant algorithms are adopted by changing certain terms from the baseline with the same data as comparison studies: without CORAL, without sigmoid entropy, without GFK, with KNN (K = 1) and with statistical features only and the GFK approach. The comparison results of the five methods are provided in Table 2.

The experiment results in Table 2 show that among all the other five approaches, the proposed approach has the best classification performance, reaching 95.56%, and the following observations can be made:(a)Effect of Domain Adaptation

As Table 2 shows, domain adaptation is an effective method to solve the different data distribution problem in transfer learning. As the results show, the accuracy of the GFK approach is the lowest with only 60.17% and the accuracy of the approach without applying GFK method is 79.90%. In this paper, the CORAL method is used to align the original signal data of both domains and the GFK method is used to align the feature vectors of two matrices. The two domain adaptation algorithms can greatly help increase the fault diagnosis accuracy.

(b)Effect of Feature Extraction

From the experiment results, the proposed hybrid feature extraction method combining the three feature extraction (95.56%) can increase the diagnosis accuracy by around 4% compared with that of the statistical feature only method (92.00%). This result shows that introducing deep features can improve the classifier performance and the proposed hybrid feature extraction method has significant influence on the incipient fault diagnosis result.

(c)Effect of Sigmoid Entropy

According to Table 2, the diagnosis accuracy of the proposed approach (with sigmoid entropy) is 95.56% which is around 20% higher than that of the approach without sigmoid entropy. Based on the experiment results, the sigmoid entropy is an effective cost function in incipient fault diagnosis.

(d)Effect of Neighbor Number

According to Table 2, the diagnosis accuracy of the proposed approach with neighbor number K = 5 in KNN is 95.56% which is around 10% higher than that of the approach with K = 1, which showed the significant influence of neighbor number on classification accuracy and the effectiveness of the proposed method.

In summary, the experiment results demonstrate that the proposed approach has achieved the best performance. Based on the results, the fundamentality of domain adaptation has been demonstrated, as both CORAL and GFK algorithms can help improve the fault diagnosis accuracy. The results also illustrate that a hybrid feature extraction method can achieve better classification performance in incipient fault diagnosis.

## 4. Conclusions

In this paper, a rolling bearing incipient fault diagnosis method based on improved transfer learning with hybrid feature extraction is proposed. Firstly, the CORAL algorithm is adopted to achieve the domain adaption of source and target domains; secondly, the proposed approach extracts features using the statistical algorithms, the wavelet scattering network and SAE network; thirdly, the proposed approach applies sigmoid entropy on extracted features of both domains; lastly, the GFK algorithm is used to align the feature vectors of both domains on the Grassmann manifold and the K-NN classifier which is trained with the source domain and is adopted to classify samples of the target domain. The experiment results show that the proposed approach has achieved highest classification accuracy among different approaches. Therefore, the effectiveness of the proposed approach in rolling bearing incipient fault diagnosis with different data distribution problem has been verified via experiments. The possible future development of this proposed approach is to reduce the computational power as several methods are adopted to extract the domain features.

## Figures and Tables

**Figure 1 sensors-21-07894-f001:**
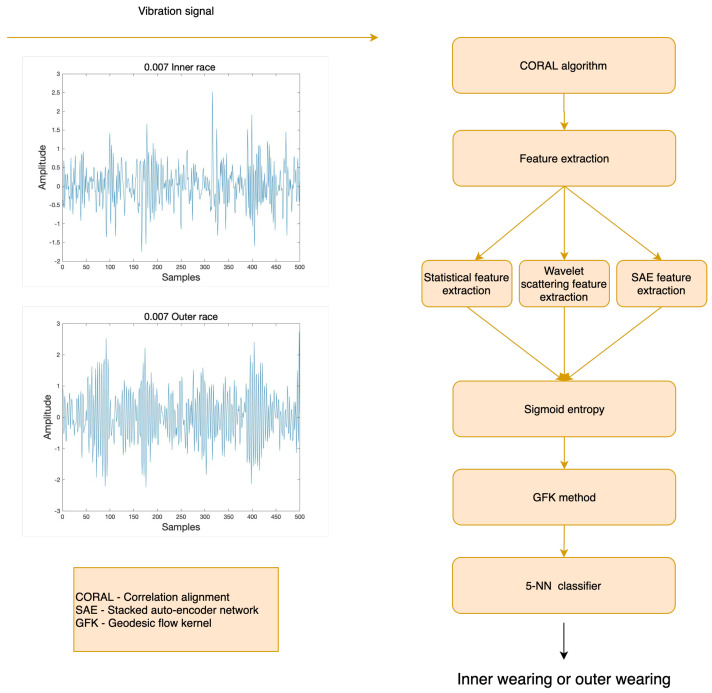
The flowchart of the proposed fault diagnosis scheme.

**Figure 2 sensors-21-07894-f002:**
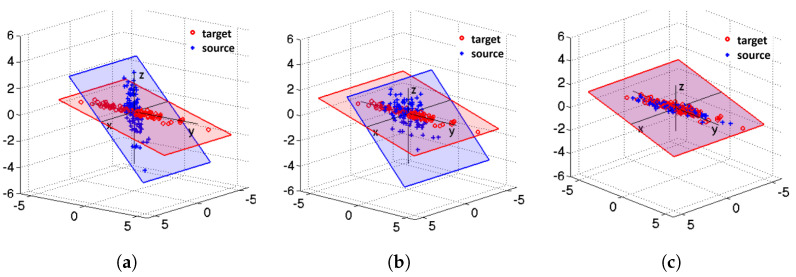
The process of CORAL algorithm: (**a**) the original data distributions of source domain and target domain; (**b**) the data distribution of source domain after decorrelation; (**c**) the data distribution of source domain after re-correlation using covariance of target domain.

**Figure 3 sensors-21-07894-f003:**
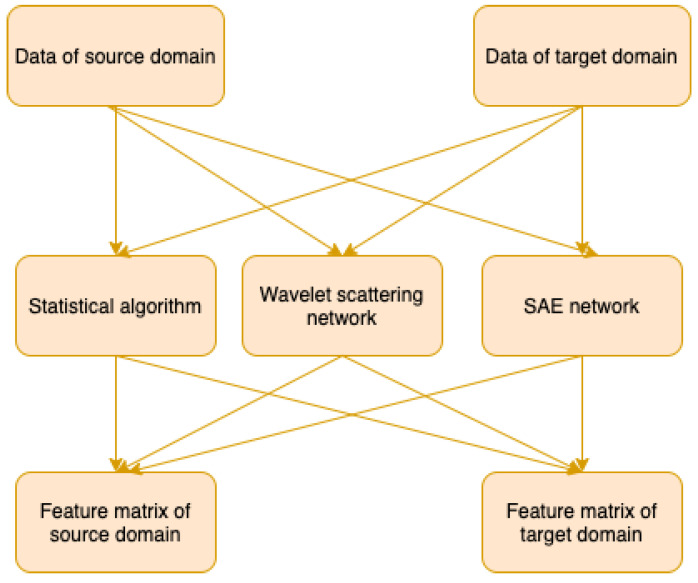
The structure of feature extraction.

**Figure 4 sensors-21-07894-f004:**
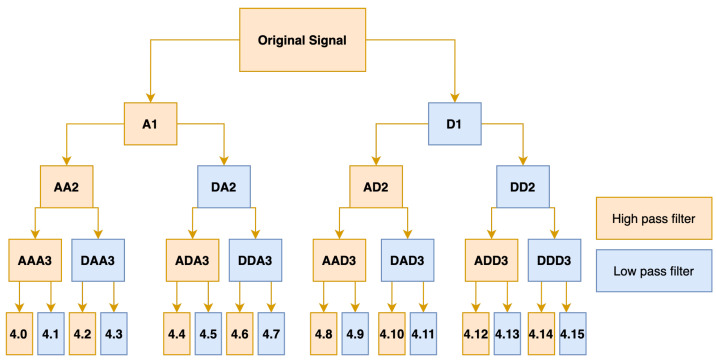
The structure of WPD.

**Figure 5 sensors-21-07894-f005:**
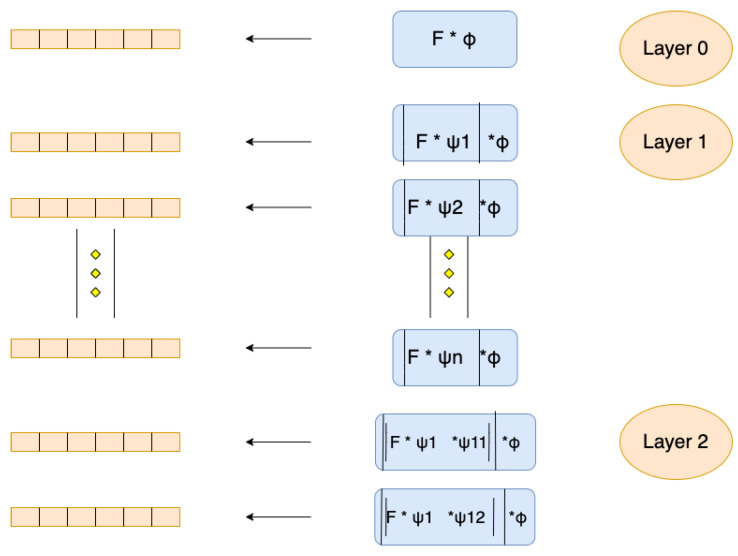
The structure of wavelet scattering network.

**Figure 6 sensors-21-07894-f006:**
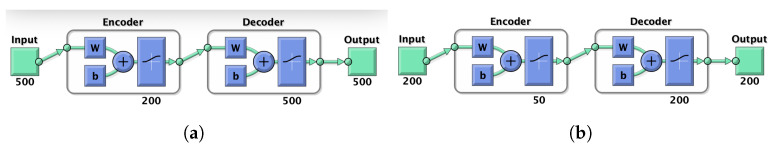
The structure of SAE: (**a**) the structure of the first SAE (the encoder output is the input of second SAE in (**b**)); (**b**) the structure of the second SAE.

**Figure 7 sensors-21-07894-f007:**
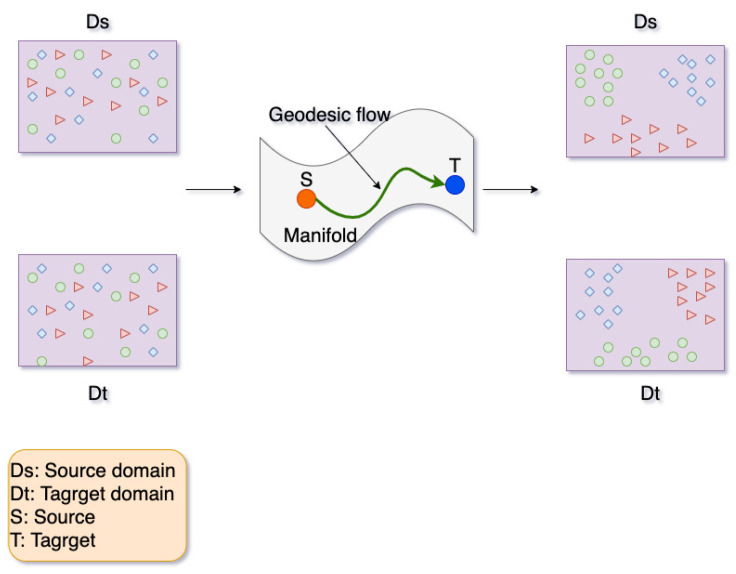
The structure of manifold learning.

**Figure 8 sensors-21-07894-f008:**
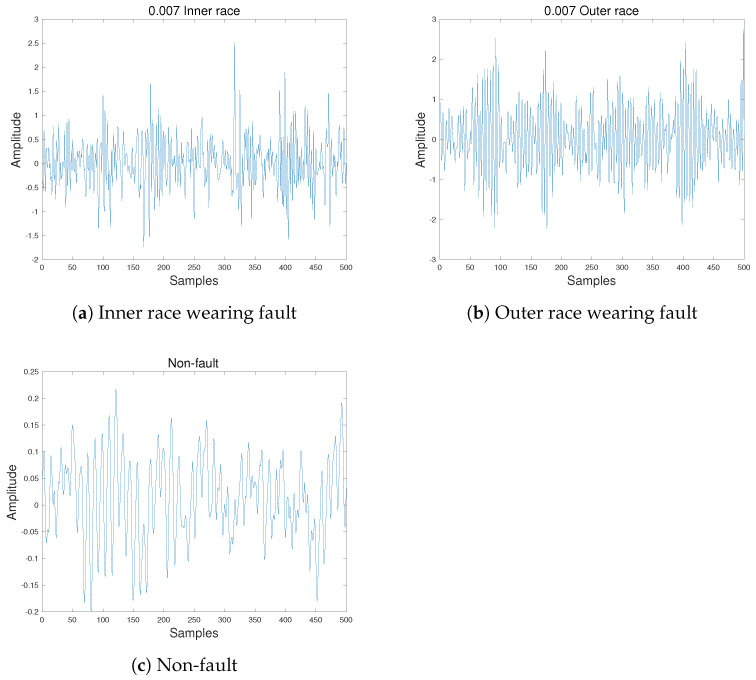
Vibration signals of two faults and non-fault in time-domain: (**a**) vibration signal of inner race wearing in time-domain; (**b**) vibration signal of outer race wearing in time-domain; (**c**) vibration signal of normal bearing in time-domain.

**Table 1 sensors-21-07894-t001:** Data of Source and Target Domains.

	Source Domain	Target Domain
Working Conditions	1797	2250
Sample Numbers	600	900
Vibration Signals in Each Sample	500	500
Fault Type	inner race and outer race wearing	Unknown
Label	1 and 2	None

**Table 2 sensors-21-07894-t002:** Experiment Results Comparison.

Approach	Source Samples	Target Samples	Accuracy
Without sigmoid entropy	600	900	75.44%
Without GFK	600	900	79.90%
KNN with K = 1	600	900	86.00%
Statistical Feature only	600	900	92.00%
GFK approach	600	900	60.17%
Proposed Approach	600	900	95.56%

## Data Availability

Restrictions apply to the availability of these data. Data was obtained from Case Western Reserve University and Xi’an Jiaotong University, and are open access from http://engineering.case.edu/bearingdatacenter/download-data-file (accessed on 16 November 2021) and http://biaowang.tech/xjtu-sy-bearing-datasets (accessed on 16 November 2021) with the permission of Case Western Reserve University and Xi’an Jiaotong University respectively.

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
