# Peer review of "Rolling Bearing Incipient Fault Diagnosis Method Based on Improved Transfer Learning with Hybrid Feature Extraction"

_sensors, 2021, doi:10.3390/s21237894_

Round 1

Reviewer 1 Report

The article presents a rolling bearing incipient fault diagnosis method based on transfer learning. The results obtained are good. However, to improve the quality of the manuscript, the following issues should be addressed:

  • The abstract and introduction should reflect the novelty of the work more clearly.
  • The authors have used CORAL algorithm in the proposed approach. Please explain why you have chosen this algorithm?
  • Tabe 1 and Table 2 compare different techniques. If possible, compare this work with other recently reported works in a table form.
  • The shortcomings of the proposed method should also be included with reasons in Conclusions section.
  • The English writing of the paper can further be improved by a thorough review from an English expert.

Reviewer 2 Report

Dear Authors,

I have read the manuscript with attention and interest. The article as a whole (composition of work, subject matter, selection of literature sources) made a decent impression on me, and during reading, I received remarks, comments, and proposed changes that I hope will serve to improve the manuscript.

In the introduction, you describe in general the various approaches and methods of feature extraction and classification. However, there is a lack of other authors' works who deal with the topic of rolling bearing incipient fault diagnosis and/or they use the mentioned approaches in their method.

In Experiments and Results/Conclusion you compare the achieved results of your own method with 5 variants of the algorithm of the method. It would be beneficial if you compared your method with other methods of other authors and compare them in terms of accuracy or other metrics e.g. (algorithm speed, computational complexity, etc.) 

Please explain how the Transfer Learning principle is used in your method.

How does your method utilize Stacked Auto-encoder Network? Stacked Auto-encoder Network for anomaly detection compares the error between the original data and its low dimensional reconstruction. What data from this step enter the step off sigmoid entropy? Reconstructed data vector or error or something else?

Formal remarks:

In all figures, please, use the same font as in the manuscript.

Fig1 vibration signal graph is not readable;

Fig2 - align the a,b,c to the middle of graphs, I suggest putting the description of a,b,c also to figure description 

Fig.4 graph of the original signal is unreadable

Fig.8,9 Please, add the references for these figures. I have found the Figure.10 in this work

https://link.springer.com/article/10.1007%2Fs10489-021-02252-2

FIG 10 I suggest to put also the graphs of non-fault signals 

L131 … feature extraction algorithms are applied…. Should there be a new sentence: … feature extraction. Algorithms are applied….? Please check the spelling.

L248 showes, Please check the spelling.

Round 2

Reviewer 1 Report

No more comments from the reviewer.

Author Response

Thanks for the valuable comment from the reviewer.

Reviewer 2 Report

Dear Authors.

Most of my comments have been included.

Just a small remark. In Fig.8,9, put the references to the end of the figure description.

Author Response

Thanks for the valuable comment from the reviewer. The original Fig. 8 and 9 have been removed to avoid the copyright issue. Both the datasets are open-access and the appropriate citations are updated accordingly.